# Mesoporous Silica and Oligo (Ethylene Glycol) Methacrylates-Based Dual-Responsive Hybrid Nanogels

**DOI:** 10.3390/nano12213835

**Published:** 2022-10-30

**Authors:** Micaela A. Macchione, Dariana Aristizábal Bedoya, Eva Rivero-Buceta, Pablo Botella, Miriam C. Strumia

**Affiliations:** 1Centro de Investigaciones y Transferencia de Villa María (CIT Villa María), CONICET-UNVM, Arturo Jauretche 1555, Villa María, Córdoba X5900LQC, Argentina; 2Departamento de Química Orgánica, Facultad de Ciencias Químicas, Universidad Nacional de Córdoba, Av. Haya de la Torre esq. Av. Medina Allende, Córdoba X5000HUA, Argentina; 3CONICET, Instituto de Investigación y Desarrollo en Ingeniería de Procesos y Química Aplicada (IPQA), Av. Velez Sárfield 1611, Córdoba X5000HUA, Argentina; 4Instituto de Tecnología Química, Universitat Politècnica de València-Consejo Superior de Investigaciones Científicas, Av. Los Naranjos s/n, 46022 Valencia, Spain

**Keywords:** hybrid nanogels, nanoarchitectonics, camptothecin, drug delivery, oligo (ethylene glycol) methacrylates

## Abstract

Polymeric-inorganic hybrid nanomaterials have emerged as novel multifunctional platforms because they combine the intrinsic characteristics of both materials with unexpected properties that arise from synergistic effects. In this work, hybrid nanogels based on mesoporous silica nanoparticles, oligo (ethylene glycol) methacrylates, and acidic moieties were developed employing ultrasound-assisted free radical precipitation/dispersion polymerization. Chemical structure was characterized by infrared spectroscopy and nuclear magnetic resonance. Hydrodynamic diameters at different temperatures were determined by dynamic light scattering, and cloud point temperatures were determined by turbidimetry. Cell viability in fibroblast (NIH 3T3) and human prostate cancer (LNCaP) cell lines were studied by a standard colorimetric assay. The synthetic approach allows covalent bonding between the organic and inorganic components. The composition of the polymeric structure of hybrid nanogels was optimized to incorporate high percentages of acidic co-monomer, maintaining homogeneous nanosized distribution, achieving appropriate volume phase transition temperature values for biomedical applications, and remarkable pH response. The cytotoxicity assays show that cell viability was above 80% even at the highest nanogel concentration. Finally, we demonstrated the successful cell inhibition when they were treated with camptothecin-loaded hybrid nanogels.

## 1. Introduction

The pharmaceutical industry constantly demands new tailored materials with enhanced properties to achieve better performance compared with the existing materials. Nanomaterials have reached a prominent position in material science for biomedical applications due to their remarkable properties, such as biocompatibility, optical behavior, excellent mechanical performance, the possibility of surface functionalization, large surface area, and tunable porosity, among others [1,2].

In this context, organic–inorganic hybrid nanomaterials, which can be also called nanoarchitectonics, formed by two or more components connected at the nanometer scale, are attractive because they combine the intrinsic properties of both materials with additional properties resulting from synergistic effects between the components and multifunctional nature [3,4]. The huge possibilities of combinations make it difficult to obtain common rules of behavior, therefore, each particular system needs to be optimized to obtain the desired properties.

Hybrid nanomaterials have several applications in nanomedicine such as wound treatment and skin regeneration [5], dentistry [6], drug delivery for a wide variety of drugs [7], and imaging-guided therapy [8,9]. The properties of this kind of hybrids can be tuned by changing the composition of materials and choosing appropriate synthetic pathways in order to obtain a tailor-made architecture with a controlled spatial distribution of both constituents leading to materials with enhanced performance characteristics, such as high thermal stability, mechanical strength, light emission, gas permeability, electron conductivity, and controlled wetting features [10].

Among the inorganic materials, mesoporous silica nanoparticles (MSNs) are interesting platforms for medical applications since they are biocompatible and exhibit effective cellular uptake, high surface area, easy pore diameter tunability, and high degree of functionalization due to the presence of a huge quantity of silanol groups on the surface, providing the possibility of loading/release of different kinds of drugs [11,12]. In addition, they have been approved by the US FDA for use in imaging diagnosis and for stage I and II clinical trials [13]. Therefore, it is expected that in a short time they can be approved as components for therapeutic formulations [14,15]. However, in many MSNs systems, drug loading is achieved by physical adsorption resulting in immediate release by diffusion after administration [16]. Thus, for the application of MSNs in drug delivery, it is essential to design hybrid materials by combining these inorganic nanoparticles with an organic component, which may act as pore gatekeeper controlling the cargo release on-demand upon application of a stimulus [17]. 

Stimuli-responsive polymers are one of the most promising alternatives as they can block the pore entrances of MSNs and upon the application of certain stimuli, they can trigger the cargo on-demand [18]. These kinds of smart materials are capable of undergoing reversible, physical, or chemical changes in their properties as a consequence of environmental variations [19]. Various external and internal stimuli can induce the conformational change and, consequently, the cargo release, including light, ultrasound, magnetic field, temperature, pH shift, redox potential gradients, ionic strength, reactive oxygen species, oxygen depletion, and unique substrates (e.g., gases, sugars, and adenosine triphosphate) [20]. Thereby, the modification of the surface of MSNs with smart polymers allows to increase the specificity of the release. 

In particular, thermo-responsive polymers exhibiting a lower critical solution temperature (LCST) in aqueous medium are promising materials due to their outstanding performance temperature-induced drug delivery or tissue engineering [21,22]. Among thermo-responsive polymers, poly(ethylene glycol) (PEG) and its derivatives exhibit thermo-sensitive nature with tunable LCST as well as antifouling properties, protein resistance (i.e., minimal protein corona formation [23]), and excellent biocompatibility [24,25]. Considering the thermo-responsive behavior, LCST is defined as the transition temperature of the conformational change in linear thermo-sensitive polymers, while the volume phase transition temperature (VPTT) is analogous for the corresponding hydrogel (or nanogel), as can be seen in Figure 1 [26]. Finally, the transition temperatures can be adjusted by combining oligo (ethylene glycol) (OEG) methacrylates of different side chain lengths; the longer the side chain, the more hydrophilic the polymer becomes [27,28].

Organic–inorganic components in hybrid nanomaterials can be assembled by physical or covalent interactions. However, it is important to highlight that covalent interactions are safer for biomedical applications because the risk of desorption of the inorganic nanoparticle in the physiological environment is minimized [29]. In addition, covalent bonding can ensure a better control of the chemical modification and distribution of specific molecules or functional groups in the structure [4]. Covalent interactions between polymers and inorganic materials can be achieved by different approaches such as “grafting-to”, “grafting-from”, etc. A particularly interesting methodology, which has been little employed for the synthesis of hybrids, is the anchoring of polymerizable groups to the surface of inorganic nanoparticles followed by polymerization in the presence of free monomer. In this case, a homogeneous distribution of inorganic nanoparticles inside the polymeric matrix is expected. For example, Hajebi et al. recently reported a smart drug-delivery system obtained by the functionalization of silica nanoparticles with 3-(trimethoxysilyl) propyl methacrylate (MEMO) followed by inverse emulsion polymerization of N,N-dimethylaminoethyl methacrylate with N,N-methylene bisacrylamide [30]. This approach allows to employ the inorganic nanoparticle as a covalent crosslinker agent in the polymerization. 

Recently, we have synthetized polymeric nanogels (PNGs) starting from OEG methacrylates, acrylic acid (AA) or itaconic acid (IA), and tetraethylene glycol dimethacrylate (TEGDMA) as crosslinker by free radical precipitation/dispersion polymerization assisted by ultrasonication, a methodology developed by our group [31]. As we had reported, this methodology ensures short reaction times, while the ultrasonication assistance allows greater control over the size of the nanogels formed. OEG methacrylates of different chain length were combined in appropriate molar radio with different percentages of acidic co-monomer to modulate the VPTT and prepare PNGs on-demand. The use of AA or IA as co-monomers improved the stabilization of colloidal particles due to electrostatic repulsion of the carboxylic acid groups and conferred pH-response to the polymeric matrix. Furthermore, these groups could be used to anchor organic molecules of interest. 

In this work, hybrid nanogels (HNGs) based on MSNs, OEG methacrylates, and acidic moieties coming from AA or IA were developed employing the same ultrasound-assisted methodology. Hence, MSNs distributed within a polymeric matrix play the role of a drug reservoir with a high cargo loading ability, inducing changes in the physicochemical properties and encapsulation/liberation of drugs, which will influence the smart behavior of the entire nanostructure. To achieve covalent anchoring between MSNs and polymer, the surface of MSNs was first functionalized with MEMO so that these functionalized structures can act as crosslinking agents. The hydrodynamic diameter (Dh), poly-dispersity index (PDI), cloud point temperature (T_CP_), zeta potential (ζ), and pH-responsiveness of HNGs were investigated by considering the effect of different compositions of the polymer structure. In addition, the effect of the degree of crosslinking on the thermo-responsive behavior was also studied, an aspect scarcely described in the literature for OEG-type nanostructures. Then, a selection of the obtained HNGs was made based on their physicochemical properties to carry out further characterization. In addition, we have also studied the loading and controlled release of camptothecin (CPT), an antitumoral drug, and cell viability in fibroblast (NIH 3T3) and human prostate adenocarcinoma (LNCaP) cell lines.

## 2. Materials and Methods

### 2.1. Reagents and Cells

The monomers di(ethylene glycol) methyl ether methacrylate (DEGMA, Sigma-Aldrich, Tokyo, Japan), oligo(ethylene glycol) methacrylate (OEGMA; Mn 475 g mol^−1^, Sigma-Aldrich, Tokyo, Japan), tetraethylene glycol dimethacrylate (TEGDMA, Sigma-Aldrich, St. Louis, MO, USA), acrylic acid (AA, Sigma-Aldrich, Steinheim, Germany), itaconic acid (IA, Sigma-Aldrich, Shanghai, China), and other reagents as N,N,N’,N’-tetramethylethylenediamine (TEMED, Sigma-Aldrich, St. Louis, MO, USA), sodium persulfate (NaPS, Sigma-Aldrich, St. Louis, MO, USA), tetraethylorthosilicate (TEOS, Sigma-Aldrich, Riedstraβe, Steinheim, Germany), hexadecyltrimethylammonium bromide (CTAB, Sigma-Aldrich, Riedstraβe, Steinheim, Germany), sodium hydroxide solution (NaOH, 1 M, Sigma-Aldrich, St. Louis, MO, USA), hydrochloric acid (HCl, Sigma-Aldrich, St. Louis, MO, USA), 3-(trimethoxysilyl) propyl methacrylate (MEMO, Sigma-Aldrich, St. Louis, MO, USA), and triethylamine (Sigma-Aldrich, Brussels, Belgium) were used as purchased.

Sodium dodecyl sulphate (SDS) was purchased from Biopack (Buenos Aires, Argentine) and camptothecin (CPT) from Fluorochem (Glossop, UK). Solvents such as toluene were supplied by Sintorgan (Buenos Aires, Argentine). Deionized purified water (Milli Q system) was employed as the polymerization medium (resistivity of 18.2 MΩ cm^−1^ at 25 °C). 

Chemicals were used as purchased except the commercial solution of AA, which was added dropwise to a column filled with silica gel in order to remove the polymerization inhibitor. 

Mouse fibroblast NIH 3T3 cell lines were obtained from American Type Cell Culture (ATCC, Rockville, MD, USA) and cultured in Dulbecco’s Modified Eagle Medium (DMEM, from Gibco, Grand Island, NY, USA) supplemented with 10% fetal bovine serum (FBS, from Lonza, Verviers, Belgium) and 1% penicillin and streptomycin (1% P/S, from Gibco) under a humidified atmosphere (5% CO_2_) at 37 °C. Human prostate cancer cell line (LNCaP) was purchased from ATCC and maintained in RPMI 1640 medium (from Gibco) supplemented with 10% FBS, 1% P/S, 2 mM L-glutamine, and 1% L-glucose (from Thermo Fisher Scientific, Waltham, MA, USA), under a humidified atmosphere (5% CO_2_) at 37 °C. In all cases, subculturing was carried out from flask to flask every 3–4 days. For this purpose, a mixture of 0.05% trypsin/EDTA solution (Sigma-Aldrich, Darmstadt, Germany) was pipetted (1 mL per 25 cm^2^) and incubated for 10 min at 37 °C. Subsequently, cells were resuspended in a small volume of fresh medium and transferred to a flask with pre-warned medium.

### 2.2. Methods

#### 2.2.1. Synthesis of Mesoporous Silica Nanoparticles

MSNs were prepared by optimizing a standard method with some modification [32,33]. Initially, 1.00 g of hexadecyltrimethylammonium bromide (CTAB) was dissolved in a mixture of water/ethanol (476 mL, 7/1). Next, 5 mL of NaOH 1 M was added to the mixture and stirred at 75 °C for a few minutes. Then, 5 mL of tetraethyl orthosilicate (TEOS) was added to the mixture and stirred for 2 h at 75 °C. The resulting mixture was cooled to room temperature, filtered off, washed with distilled water and methanol, and dried. In order to remove the template and generate the mesopores, the filtrate was refluxed in 150 mL of HCl solution (0.25 N) for 24 h. The white solid was filtered and subjected to another cycle of reflux under the same conditions for 24 h. Finally, the solid was filtered, washed with distilled water and ethanol, and dried overnight at room temperature under vacuum.

#### 2.2.2. Surface Modification of Mesoporous Silica Nanoparticles

The surface functionalization of the MSNs consists in a silanization by using 3-(trimethoxysilyl)propyl methacrylate (MEMO). Briefly, 25 mg of MSNs were suspended in 12.5 mL of toluene. Then, 625 μL of triethylamine, 125 μL of MEMO, and 25 μL of water were added, and the resulting dispersion was stirred for 24 h of reaction at room temperature. The reaction mixture was purified by centrifugation and washed 3 times; the first time with acetone, and then with ethanol. Finally, MSN@MEMO were resuspended in ethanol.

#### 2.2.3. Synthesis of Hybrid Nanogels

The synthesis of dual-responsive HNGs was carried out following a previous protocol developed in our group, which consists of a free radical precipitation/dispersion polymerization assisted by ultrasonication [31,34]. First, the OEG monomers (DEGMA and OEGMA, 1 mmol), the acid co-monomers (AA or IA), MSN@MEMO (0.25 mg), and sodium dodecyl sulfate (SDS, 0.6 mM) were placed in a flask. Finally, solutions of the TEMED (0.075 mmol) and NaPS (0.015 mmol) employed as catalyst and initiator, respectively, were injected with the amount of Milli Q water necessary to complete a final volume of 10.0 mL. Then, the reaction mixture was ultrasonicated discontinuously at 70 °C: 1 min pulse sonication at 70% power in an Omni Ruptor 4000 ultrasonication tip probe instrument, 1.5 min of stirring to complete a total of 6 pulses of ultrasound and 15 min of reaction.

Different experimental conditions were explored such as DEGMA:OEGMA ratio, amount of acidic co-monomer, incorporation of polymeric crosslinker, etc. (See Table 1).

For each composition, three samples resulting from three different syntheses were obtained.

The HNGs obtained were purified by dialysis against water for 4 days at room temperature employing a 50 kDa MWCO membrane. Subsequently, a portion of the samples were freeze-dried to their characterization.

#### 2.2.4. Characterization of Hybrid Nanogels

Mesoporous silica nanoparticles size distribution (hydrodynamic diameters, Dh) and ζ-potential of as-prepared MSNs were determined by dynamic light scattering (DLS) and electrophoretic light scattering (ELS) using Zetasizer Nano ZS (Malvern Instruments Ltd., Worcestershire, United Kingdom). Powder Xray diffraction (XRD) patterns were collected in a Philips X’Pert diffractometer equipped with a graphite monochromator, operating at 40 kV and 45 mA. Nanoparticle morphology was investigated by transmission electron microscopy (TEM) in a JEOL JEM 2100F microscope operating at 200 kV. Nitrogen gas adsorption isotherms were measured in a Micromeritics Flowsorb apparatus. Surface area calculations were performed using the Brunauer−Emmett−Teller (BET) method [35], whereas pore size distribution was calculated according to the Kruk−Jaroniec−Sayari estimation [36]. 

The HNGs were characterized by Fourier transform infrared (FT-IR), measured in a Nicolet iN10 (Thermo Scientific, Waltham, MA, USA; ST 2425-CONICET) instrument using a liquid nitrogen cooled mercury cadmium tellurium (MCT) detector. Nuclear magnetic resonance (NMR) spectra were carried out in a Bruker Ultra Shield 400 (^1^H-NMR: 400 MHz ST 1027-CONICET) in deuterated water (D_2_O). Moreover, micrographs of the HNGs were performance by transmission electron microscopy (TEM) in a JEOL JEM EXII 1200 operating with 80 kV (INTI-IPAVE-CONICET). Dh and ζ-potential were determined by DLS and ELS using a Zetasizer Nano ZS (Malvern Instruments Ltd., Worcestershire, UK) with a He–Ne laser (l ¼ 633 nm) and scattering angle of 173° instrument. The informed Dh is the average size obtained from three samples from different syntheses under the same conditions. Furthermore, the standard deviation (SD) is obtained by considering the average Dh of similar samples from different synthetic procedures.

#### 2.2.5. Thermo-Responsive Behavior

The effect of temperature was evaluated both by DLS and by turbidimetry. The turbidimetry technique was used to determine the T_CP_ by using a UV-Visible spectrophotometer equipped with temperature control. Spectra of an aqueous dilution of obtained HNGs was recorded every 2 °C in a temperature range from 15 °C to 50 °C, or higher when it was necessary. T_CP_ values were determined by fitting a Boltzmann’s basic sigmoid function on the plot of % transmittance against temperature (for a λ of 670 nm).

For the DLS analysis, VPTT can be determined by measuring each sample of HNGs in the temperature range from 25 °C to 60 °C. In this case, 100 μL of each sample of HNGs were re-suspended in 1 mL of Milli Q water grade.

#### 2.2.6. pH-Responsiveness

The behavior of the HNGs with different amounts of IA were determined against pH. The effect of pH was evaluated by determining the values of Dh in aqueous solutions of different pH values (3, 5, 7, and 9). The samples were prepared by adding 150 µL of the HNGs suspension in 1.5 mL of the corresponding pH medium. These solutions were prepared using NaOH and HCl 0.1 M. The dispersion obtained was allowed to stabilize overnight. The pH of the dispersion was determined just before measuring and if necessary, the pH was adjusted to its desired value. Finally, the dispersion was sonicated for 5 min and the Dh values were determined.

#### 2.2.7. Drug Encapsulation

A stock solution (3.0 mg/mL) of camptothecin (CPT) in a mixture of methanol and hydrochloric acid (MeOH:HCl 95:5) was prepared. Then, 0.5 mL of this stock solution of CPT, 0.3 mL of hybrid nanogel HNG-P(DEGMA-co-IA_12_) (4.4 mg/mL), and 0.2 mL of water were added in 1.5 mL Eppendorf tubes and incubated at room temperature and shaken at 1500 rpm in a thermomixer for 24 h. Subsequently, the tubes were centrifuged (13,200 rpm, 10 min, 4 °C) and the supernatants were retired. Every residue was washed with 1 mL of Milli Q water grade. Afterwards, aliquots were centrifuged (13,200 rpm, 10 min, 4 °C) and the supernatants were retired. 

To carry out the quantification of the CPT, the samples were reconstituted with 1 mL of a mixture of MeOH:HCl 95:5 (*v/v*) and analyzed by UV-Visible spectrophotometry; the absorption was measured at 302 nm to calculate the efficiency of encapsulation and loading of the drug [37,38,39]. For this, a calibration curve was previously performed to obtain the molar absorptivity coefficient (see Appendix A). 

#### 2.2.8. Drug Release

Studies of CPT release from HNGs as a function of temperature were performed at 40 °C. For this, Franz diffusion cells with a 50KD cellulose membrane were used. The previously lyophilized NG pellet was reconstituted in 1.0 mL of phosphate buffered saline (PBS, pH 7.4), this suspension was placed in the donor compartment of the cell and 4.0 mL of medium (PBS (pH 7.4)) were placed in the recipient compartment. At predetermined time intervals (0.25, 0.5, 1.0, 2.0, 3.0, 6.0, 8.0, 24.0, and 48.0 h), 1.0 mL aliquots were taken from the receptor compartment and replaced with an equal amount of fresh medium. These experiments were carried out in a thermostatic bath maintaining constant agitation (120 rpm) and temperature (40 °C). Collected aliquots were lyophilized and reconstituted in 1.0 mL of MeOH:HCl to determine the amount of CPT released by UV-Visible spectrophotometry at 302 nm. Each experiment was performed in triplicate.

#### 2.2.9. Cell Viability Assay

The effect of different materials on cell metabolic activity was determined using the MTS (3-(4,5-dimethylthiazol-2-yl)-5-(3-carboxymethoxyphenyl)-2-(4-sulfophenyl)-2H-tetrazolium) colorimetric assay. Cells were seeded in a 96-well plate at a density of 4000 (NIH 3T3) or 10,000 (LNCaP) cells per well and cultured in 5% CO_2_ at 37 °C for 24 h. Then, cells were treated with the naked material (HNG-P(DEGMA-co-IA_12_), CPT-loaded HNG-P(DEGMA-co-IA_12_) or free CPT (stock solution in DMSO), with final doses ranging from 2.5 to 0.0005 µg mL^−1^ during 72 h. At the end of the incubation period, 15 μL of MTS solution was added into each well and incubated for another 4 h. Absorbance was measured with a Perkin Elmer Wallac 1420 VICTOR2 Multilabel HTS Counter (Northwolk, CT, USA) at the wavelength of 490 nm. IC_50_ survival data were calculated by non-linear regression sigmoidal dose-response (variable slope) curve-fitting using Prism 6.0 software (GraphPad, San Diego, CA, USA). Three independent experiments were performed for every sample and each experiment was carried out by in triplicate.

## 3. Results and Discussion

### 3.1. Synthesis and Structural Aspects of Hybrid Nanogels

Hybrid nanomaterials based on MSNs and stimuli-responsive polymers are excellent candidates for the transport and release of therapeutic agents due to the well-defined tunable physicochemical properties and porous structure of MSNs, and the improved efficiency that confers the organic component. The planification of the synthesis methodology is essential for obtaining the desired morphology, architecture, and properties in the final nanostructure.

In the present work, the synthetic strategy allows covalent incorporation of MSNs into the polymer matrix to form HNGs. Figure 2 shows the general procedure for the synthesis of these hybrid nanostructures. It consists of two steps: the silanization reaction of the MSNs (1) and the polymerization of the functionalized MSNs (HNG-P(DEGMA)) (2).

Firstly, MSNs were prepared by using a standard method using TEOS and CTAB, which allowed to obtain MSNs with regular sizes and spherical mesoporous shape with an average diameter of about 114.09 ± 17.71 nm (Appendix A). Dh and ζ-potential were also determined (Appendix A). The mesoporous structure of MSNs was determined by Nitrogen adsorption–desorption isotherms, which show the typical IV isotherm, which confirms the mesoporous nature of MSNs, with a high surface area (approximately 1100 m^2^ g^−1^) and a pore volume of 1.02 cm^3^ g^−1^ (Appendix A) [40]. Moreover, these nanoparticles showed the typical X-ray pattern with an intense diffraction peak (100) at 2θ value of 2.21 (Appendix A) [32].

To achieve the covalent incorporation of MSNs to the organic matrix, MSNs were first functionalized with vinyl groups which can then be polymerized by radical polymerization. Hence, this functionalization allows that MSNs can act as crosslinkers in the synthesis of the polymeric material. Thus, the first step of the synthesis is the functionalization of MSNs surface with a 3-(trimethoxysilyl) propyl methacrylate (MEMO) in order to anchor the polymerizable groups onto the inorganic component. The conditions of the silanization reaction were optimized and set under continuous stirring in toluene at room temperature for 24 h of reaction. As a parameter to evaluate the degree of silanization of MSNs in each sample, we used a relation of absorbances in the FT-IR spectra. The relation results from the maximum absorbances values from a band of MEMO corresponding to C=O (around 1720 cm^−1^) and a band of MSNs attributed to Si-O-Si (around 1100 cm^−1^). These optimization studies and further details are shown in the Appendix A.

For the synthesis of the smart polymeric material, we have used DEGMA and OEGMA as the thermo-responsive monomers. According to literature, the LCST of the linear homopolymer P(DEGMA) was determined to be 26 °C, whereas linear polymer P(OEGMA) (*n* ∼ 9) exhibits LCST values around 90 °C [41]. Moreover, AA or IA were used as pH-responsive co-monomers and TEGDMA as the crosslinker. Ultrasonication-assisted free radical precipitation/dispersion polymerization was employed as the synthetic procedure. As indicated in a previous work, in the presence of inorganic NPs, the ultrasonication strategy should guarantee their colloidal stability during polymerization [42]. Therefore, inorganic cores are expected to be homogeneously distributed inside the polymeric network, avoiding particle aggregation, and maximizing the available surface for the polymeric covering. 

FT-IR studies were performed to analyze the composition of the resulting HNGs (Figure 3). The FT-IR spectrum of the MSNs used (blue line) shows around 1100 cm^−1^ the signal corresponding to the vibration of the Si-O-Si (st) bond. The band at approximately 3400 cm^−1^ and the peaks around 1630 cm^−1^, 960 cm^−1^, and 800 cm^−1^ can be assigned to the Si-OH groups on the surface [43,44]. 

The functionalization of MSNs with MEMO can be confirmed by the appearance of the distinctive signal of vinyl groups C=C (between 1690 and 1635 cm^−1^) [45]. In addition, the characteristic bands corresponding to the aliphatic stretch vibration of C–H in the range 3000–2840 cm^−1^, the stretch of the esters corresponding to C=O at 1718 cm^−1^ and C-O-C vibration between 1050 and 1330 cm^−1^ can be observed [46]. 

After polymerization with DEGMA, the synthesized hybrid NGs also show the characteristic bands corresponding to the aliphatic stretch vibration of C–H in the range 3000–2840 cm^−1^, the stretch of the esters corresponding to C=O at 1732 cm^−1^ and C-O-C between 1050 and 1330 cm^−1^. In addition, vibration of O-CH_3_ of P(DEGMA) at around 2825 cm^−1^ can be observed, as previously reported [47]. Furthermore, when acidic co-monomers are incorporated, their presence in the polymeric matrix can be confirmed by the analysis of the carbonyl stretching FT-IR bands (see Appendix A).

H^1^ NMR spectra were also analyzed to confirm the silanization of MSNs and the formation of P(DEGMA) (see Appendix A). 

### 3.2. Characterization of Hydrodynamic Diameters and Dual Smart Responsiveness

The synthetic variables such as the nature of monomers, molar radio of monomers, concentration of polymer, and degree of crosslinking, can influence the physicochemical properties of NGs and their thermo-responsive behavior. In the following sections, the results obtained by measurements of DLS and turbidimetry of aqueous suspensions for HNGs are discussed. The average hydrodynamic diameter (Dh) comes from the DLS intensity distribution. 

Regarding the thermo-responsive behavior, polymers made of OEG side chains are completely miscible with solvent and become insoluble upon heating. Consequently, the phase transition takes place at the LCST [27]. Therefore, an aqueous suspension of this kind of polymers is clear and homogeneous below LCST. Conversely, above the LCST, the polymer becomes hydrophobic and water insoluble so the suspension gets cloudy [48,49]. Thus, this macroscopic change can be monitored by turbidimetry allowing the determination of a transition temperature called cloud point temperature (T_CP_), which is an important parameter to determine. Furthermore, VPTT and the pH-sensitivity of these HNGs can be evaluated considering Dh as a function of the temperature or pH of media, respectively.

#### 3.2.1. Effect of Molar Ratio of OEG Monomers and Crosslinking

Our previous studies performed with PNGs and magnetic NGs, demonstrated that the molar ratio of OEG monomers: DEGMA:OEGMA 80:20 allows to obtain appropriate LSCT for biomedical applications [31,34,42]. Therefore, in the present work, this condition was explored with the inorganic MSNs. In addition, despite the fact that MSN@MEMO can act as a crosslinker itself, the incorporation of additional crosslinker (TEGDMA) was explored. Therefore, Table 2 shows the effect of the degree of crosslinking (with/without TEGDMA) and the effect of the incorporation of acidic co-monomer (AA) in the Dh and thermo-responsiveness, keeping the amount of MSN@MEMO constant.

Starting from HNGs with DEGMA:OEGMA 80:20 and 1.5% of TEGDMA co-monomer, nanosized particles with monodisperse distribution are obtained (HNG-P(DEGMA-co-OEGMA)^1.5^). When their size is compared to their analogous HNG with DEGMA:OEGMA 80:20 without the TEGDMA co-monomer but with the same amount of MSN@MEMO, it can be observed that HNGs without the polymeric crosslinker are slightly smaller than the former, but the difference is really negligible. However, the PDI results higher for the less crosslinked HNG, without the polymeric crosslinker. The more crosslinked system would result in stiffer structures, in which the closeness between chains would increase the physical interactions, thus contributing to the uniformity of sizes. 

The T_CP_ obtained for HNG with DEGMA:OEGMA 80:20 and 1.5% of TEGDMA co-monomer is too high (59.5 °C) considering biomedical applications. This can be explained by considering the crosslinked structure due to the presence of both TEGDMA and MSN@MEMO, which demands more energy to lead to the conformational swelling change. Otherwise, the matrix without adding TEGMA crosslinker exhibits a T_CP_ of 46.6 °C. Therefore, when only the inorganic structure is used as crosslinker, the T_CP_ results more than 10 °C lower. Similar behavior was observed in PNGs with DEGMA, OEGMA, TEGDMA, and AA or IA (without inorganic NPs): the stiffer the NG structure, the higher T_CP_ [34]. In poly-N-isopropylacrylamide (PNIPAM) micro/nanogels with and without inorganic NPs, other authors have noticed that the swellability decreases with the increase of the crosslinker content [50,51,52,53,54]. Moreover, París and coworkers observed that in P(DEGMA) hydrogels using TEGDMA as crosslinking agent, the VPTT became higher as the degree of crosslinking of the hydrogels increased [55]. 

In view of these results, in the next section, compositions with MSN@MEMO as the only crosslinker were developed and evaluated for thermo-sensitive behavior and other properties. Furthermore, the high T_CP_ values observed when the polymer is composed of DEGMA:OEGMA 80:20 indicate that, in comparison with PNGs, the presence of MSNs induce an increment in this parameter. This may be related with the hydrophilic nature of the MSNs and the stiffer structure of HNGs compared with PNGs. Therefore, on account of the high T_CP_ values, HNGs with constant MSN@MEMO and only DEGMA as the only thermo-sensitive monomer were studied. In addition, 4% of AA or IA was incorporated.

#### 3.2.2. Effect of Incorporation of AA and IA

Data obtained by DLS measurements and turbidimetry of aqueous suspensions for HNGs with organic component P(DEGMA) and with 4% of acidic co-monomers (AA and IA) are listed in Table 3.

In the absence of acidic co-monomer, HNG-P(DEGMA) exhibits an average size of 206.9 nm with an associated SD of 103.6 nm. This high value of SD indicates some degree of irreproducibility between the syntheses performed. With respect to the thermo-response of these HNGs, a T_CP_ value of 23.6 °C was observed. Thus, when compared with HNG-P(DEGMA-co-OEGMA), it was possible to achieve a decrease in T_CP_ value of 23 °C due to a polymeric composition based on only the shorter methacrylate, which leads to a more hydrophobic structure with a lower T_CP_ value [27,56]. This result represents a good starting point for the study of the incorporation of acidic co-monomers, AA or IA. 

The synthesis with acidic moieties (AA or IA, 4%) allow to obtain low polydispersity distributions of nanosized HNGs. Furthermore, lower values of SD are observed in these cases, indicating more reproducible results between batches. According to this, it seems that the incorporation of acidic moieties improves reproducibility, which may be related to the enhanced stabilization of colloidal particles through the repulsion of negative charges of carboxylic groups. 

The Dh of HNG-P(DEGMA-co-AA_4_) is similar to HNG-P(DEGMA) (237.4 vs. 206.9 nm), whereas HNG-P(DEGMA-co-AI_4_) results smaller (137.5 nm). The presence of ionizable groups in the structure of polymeric NGs can induce a greater ability to stabilize precursor particles in the synthetic process, which may lead to smaller nanostructures. Nonetheless, this size reduction is only observed for the case of the IA. 

Regarding the thermo-responsive behavior of these HNGs, the incorporation of 4% of acidic co-monomers produces an increment in the T_CP_ values. This can be explained taking into account the hydrophilic nature of acidic co-monomers. Nonetheless, in the case of 4% of AA, the T_CP_ results much higher than for 4% of IA (34.6 vs. 28.0 °C). Similar observations were made for PNGs based on OEGMA, DEGMA, and AA or IA previously reported [34]. In aqueous suspensions, it is reasonable to expect that one of the carboxylic groups of IA molecule is protonated. Thus, in these conditions, it seems that acidic groups of IA form intramolecular hydrogen bondings which act as hydrophobic arrangements in which polymer–polymer interactions prevail over the polymer–water ones. This hydrophobic behavior was previously reported in other thermo-responsive polymeric materials [57,58,59,60,61,62]. In the present case, these polymer interactions lead to a lower T_CP_ value than expected.

This interesting behavior of IA allows us to propose the incorporation of greater amounts of this acidic co-monomer before reaching T_CP_ values close to the body temperature needed for biomedical applications. 

Hence, considering these results, the next syntheses were performed using constant amount of MSN@MEMO, DEGMA, and rising the IA content. 

#### 3.2.3. Effect of Increasing IA Concentration

Table 4 shows the results obtained by DLS measurements and turbidimetry of aqueous suspensions for HNGs based on constant amount of MSN@MEMO, DEGMA, and different compositions of IA. 

From Table 4 it can be seen that all the samples show a size distribution in the nanometric scale with polydispersity index (PDI) values between 0.1 and 0.4. 

Analyzing the NGs with variable percentage of IA, a general trend can be observed: as the IA concentration is increased, the size of HNGs gets higher according to a rise of their swelling grade because the hydrophilicity of the final matrix is higher [41]. In addition, the corresponding PDI values also become higher, indicating an increase of the polydispersity of the distributions. 

Comparing the results of DLS measurements of previously developed PNGs based on p(DEGMA-co-OEGMA-co-IA) without MSN component, it can be observed that PNGs with DEGMA: OEGMA 80:20, 4% IA, and 1.5% of TEGDMA crosslinker exhibit much larger Dh at room temperature (345.2 nm) than HNG-P(DEGMA-co-IA_4_). Therefore, by using DEGMA and MSN@MEMO as crosslinker agents, it was possible to reduce Dh parameter. Furthermore, it is important to highlight that it was possible to increase IA percentage to 12% maintaining the size of the HNGs at the nanoscale, contrary to what was observed in PNGs [34].

ζ-potential was negative in all samples, and its absolute value increased with the concentration of IA as expected due to the incorporation of carboxylic groups, which can be unprotonated giving negative charges. 

In order to determine thermo-responsive behavior, hydrodynamic diameters were also measured at higher temperature (50 °C). Data show that HNG-P(DEGMA) and HNG-P(DEGMA-co-IA_4_) exhibit higher sizes, whereas HNG-P(DEGMA-co-IA_8_) and HNG-P(DEGMA-co-IA_12_) show shrunken matrices at 50 °C. The higher size of HNG-P(DEGMA) may probably be related to an agglomeration process of HNGs driven by the increase in temperature. This behavior is expected because these HNGs are poorly charged, and therefore are more susceptible to agglomeration. With 4% of IA content, measurements showed a slight increase in size at 50 °C, but considering the SD values, this increase is not significant. Otherwise, HNGs with more IA content (8 and 12%) exhibit significant lower sizes at 50 °C compared with room temperature, indicating the shrinkage of the matrices with the temperature increase. Therefore, it can be stated that the contraction of these HNGs can be observed at IA compositions greater than 8%.

Last column of Table 4 shows the T_CP_ values obtained by turbidimetry. HNG-P(DEGMA) exhibits the lower T_CP_ value, while afterwards a general trend can be observed: the T_CP_ values increase when the amount of IA gets higher due to the increase of hydrophilicity [27]. 

#### 3.2.4. Characterization of the pH-Responsiveness

The pH-responsiveness of HNGs with IA was determined by measuring the hydrodynamic particle diameter (at 25 °C) as a function of pH (Figure 4). HNG-P(DEGMA) was not sensitive to the pH as expected because it does not have carboxylic acid groups in its polymeric matrix (Figure 4a). 

The other HNGs with increasing percentage of IA display pH-behavior since their sizes increase with the rise of the pH of the environment. This pH-sensitivity is a consequence of the presence of carboxylic groups, which when deprotonated produce the electrostatic repulsion between the negative charges, leading to the swelling of the polymeric material. 

The measurements of DLS of HNG-P(DEGMA-co-IA_4_) and HNG-P(DEGMA-co-IA_8_) with 4 and 8% of IA, respectively, show agglomerated particles and polydisperse distribution at pH 3, which indicate that NGs are interacting with each other (data not shown). Therefore, in Figure 4b, we show the sizes obtained at pH 5, 7, and 9. 

The response as a function of pH results more evident when the amount of IA is greater. Consequently, the change in size between pH 5 and 9 is 131.5, 164.5, and 422.3 nm for HNG with 4, 8, and 12% of IA, respectively. Therefore, HNG-P(DEGMA-co-IA_12_) is the one that exhibits more significant pH response. 

For all the above results, it results clear that HNG-P(DEGMA-co-IA_12_) with the greater amount of IA tested (12%) exhibits an excellent performance in the desired dual response, pH, and temperature. Despite the high content of hydrophilic monomer, the obtained size is appropriated for drug delivery applications. 

### 3.3. Further Characterization of the Selected Hybrid Nanogel, Camptothecin Encapsulation and Release Studies

Figure 5a illustrates the macroscopic change that can be observed when suspensions of HNGs are heated. As it was mentioned above, the increase in cloudiness can be measured by turbidimetry. 

The plot of average Dh vs. temperature of HNG-P(DEGMA-co-IA_12_) (12% IA) is represented in Figure 5b. The data, which fit Boltzmann’s sigmoidal equation, show that the temperature at which the first derivative of the function is maximum results 33.1 °C, which represents the VPTT value. This result does not match with the T_CP_ value obtained by turbidimetry (37.2 °C); however, this difference can take place because each method evaluates different parameters. 

TEM image (Figure 6a) shows the nature of HNG-P(DEGMA-co-IA_12_) nanoparticles, with little polydispersity and quite narrow particle size distribution (e.g., 80–350 nm), and average diameter of about 170.2 ± 45.3 nm (Appendix A). Indeed, by this technique very little aggregates were observed. However, the observed Dh obtained by DLS (Table 4) is two-fold the TEM average value, which can be explained by considering the swelling of the crosslinked networks in solution. Moreover, the distribution of the inorganic and organic component within the hybrid nanomaterial can be also evaluated by TEM (Figure 6b, Appendix A). Images show that each HNG is formed by 1, 2, or 3 MSNs inside a polymeric matrix. 

With the objective of developing a new novel hybrid drug delivery system, the hydrophobic molecule, CPT, was selected as a model antitumor drug to evaluate the transport ability of these nanocarriers. This drug, which exhibits a water solubility of 0.511 mg/mL, was loaded into these nanogels to increase its bioavailability and prevent its rapid plasma clearance, high systemic toxicity, and poor selectivity towards cancer cells. Due to the chemical nature of CPT, the main intermolecular interactions between CPT and polymeric network are Van der Waals forces.

Drug encapsulation assays allow to obtain remarkable high encapsulation efficiency and drug loading content. Moreover, SD values indicate highly reproducible protocol.

Encapsulation Efficiency = (94 ± 2)%Drug loading content = (52.0 ± 0.7)%

These values are considerably higher than those that can be found in literature for similar systems [38,39,63]. This favorable fact could be due to the nature of the hybrid system, in which the porous structure of MSNs allows a higher incorporation of the hydrophobic drug compared to other nanocarriers. CPT release was performed in PBS pH 7.4 and 40 °C as a model experiment to ensure physiological conditions and a temperature above the VPTT of these HNGs (Figure 6c). Considering the calibration curve obtained (Appendix A), the final CPT concentration is above 30 μM, which, due to the extreme cytotoxicity of this drug, is enough to achieve a therapeutic effect [64]. 

The release mechanism of CPT from HNG-P(DEGMA-co-IA_12_) nanoparticles is very difficult to figure out, as a results of its hybrid nature. In this context, we have fitted the experimental data into a standard kinetic model, specifically, a first-order equation (Equation (1)) [65], the Higuchi equation (Equation (2)) [66], and the Bhaskar equation (Equation (3)) [67]: (1)$$X=1-e^{-k\cdot t}$$
(2)$$X=k\cdot t^{0.5}$$
(3)$$X=1-e^{-k\cdot t^{0.65}}$$
where X, t, and k are, respectively, the fraction of discharged drug, release time, and the kinetic constant. Here, the comparison of fitting results and the obtained k and r values is presented in Appendix A. It is noticeable that the release data follow very well the first-order equation, that is, the release rate depends exclusively in the remaining concentration of the drug, with little, if any, restrictions by diffusion through the polymeric matrix.

### 3.4. Cytotoxicity Study

CPT is a wide-spectrum antitumoral drug, which has been tested at the preclinical stage mostly for the therapy of colorectal, stomach, neck, bladder, breast, lung carcinomas, and leukemia [64]. Unfortunately, its clinical used is hampered due to the extreme toxicity, lower biodisponibility, and the unstability of the lactone ring in physiological conditions. For this purpose, it is imperative to develop drug delivery systems able to selectively deliver the therapeutic payload to the cancer cells also imposing a controlled release under the action of specific stimuli. Here, as shown in Figure 7, the cytotoxicity of CPT and CPT-loaded hybrid nanogel was determined by MTS assay at different concentrations over NIH 3T3 and LNCaP cells. Firstly, we tested the biocompatibility of the nacked material at the corresponding nanoparticle concentration range. This material was evaluated in a separate experiment, showing that the relative cell viability was above 80% even at the highest HNG-P(DEGMA-co-IA_12_) concentration, corresponding to a good biocompatibility profile (see Appendix A). 

Significant dose-dependent cytotoxicity was observed in both cases for LNCaP and NIH 3T3 cells. The IC_50_ for CPT and the hybrid nanogel was similar in LNCaP cells. In NIH 3T3 cells, the results showed that the hybrid nanogel is a bit less active than the free drug (Table 5). These results suggest that the hybrid nanogel under the microenvironment of cancer cells improves the release of CPT, which causes cell inhibition. However, the incorporation of the antitumor drug as a cargo into the hybrid nanoparticles ensures a “zero-release” of CPT before the uptake by cancer cells, as this will be promoted under pH-stimulus (e.g., lower pH at the lysosomes [68]). Furthermore, the organic and hydrophobic nature of HNGs polymer at temperatures above VPTT also favors cell internalization by clathrin-dependent endocytosis [69], which accelerates the uptake. 

## 4. Conclusions

We reported a synthetic approach to produce uniform distributions of HNGs based on MSNs, OEG methacrylates and acidic co-monomers. The chosen strategy allows covalent bonding between the inorganic component and the polymer by driving a prior functionalization of MSNs in order to use them as covalent crosslinkers. The resulting HNGs are colloidally stable, which may be related to the ionic stabilization given by carboxylic acids. We have carried on an optimization of the polymeric composition of HNGs: degree of crosslinking, molar radio of DEGMA: OEGMA, nature of the acidic co-monomer, and percentage of acidic co-monomer in the synthesis. The optimal composition consisted in 0.25 mg of MSN@MEMO, DEGMA as the only thermo-responsive monomer (1 mmol), and 12% of IA. This composition and, particularly, the use of MSNs as crosslinkers allows the incorporation of higher percentages of IA comparing to previous reports, maintaining nanosized homogeneous distribution and higher stability. 

The designed HNGs exhibit dual smart behaviour. On the one hand, the HNG-P(DEGMA-co-IA^12^) display a T_CP_ of 37.2 °C, which is a desirable value for biomedical applications. On the other hand, this HNGs show a notable pH response, starting from around 200 nm at low pH and achieving 650 nm at pH of 9. HNGs with lower amounts of IA display lower size change with the pH of media.

Moreover, CPT was encapsulated into the HNGs, obtaining remarkable high encapsulation efficiency values compared with the literature. We attribute this to the mesoporous structure of the inorganic component, which acts as a reservoir of the drug [70].

The cytotoxicity assays show that cell viability was above 80% even at the highest HNGs concentration. Finally, we demonstrated the successful cell inhibition when they were treated with loaded HNGs. Overall, the combination of both components, an inorganic reservoir core and an organic stimuli-responsive coating, allows to obtain a novel drug delivery system capable of producing an efficient controlled release of CPT in cancer cells.

## Figures and Tables

**Figure 1 nanomaterials-12-03835-f001:**
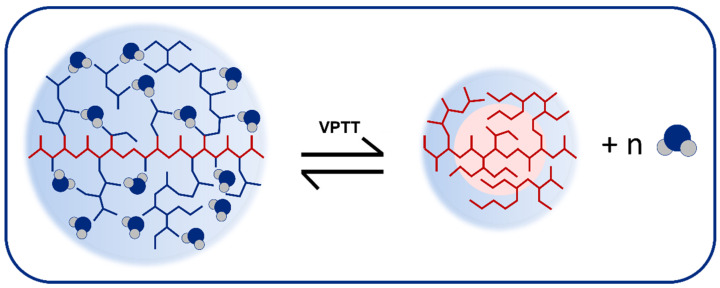
A schematic representation of the thermo-response of this type of polymeric nanogels.

**Figure 2 nanomaterials-12-03835-f002:**
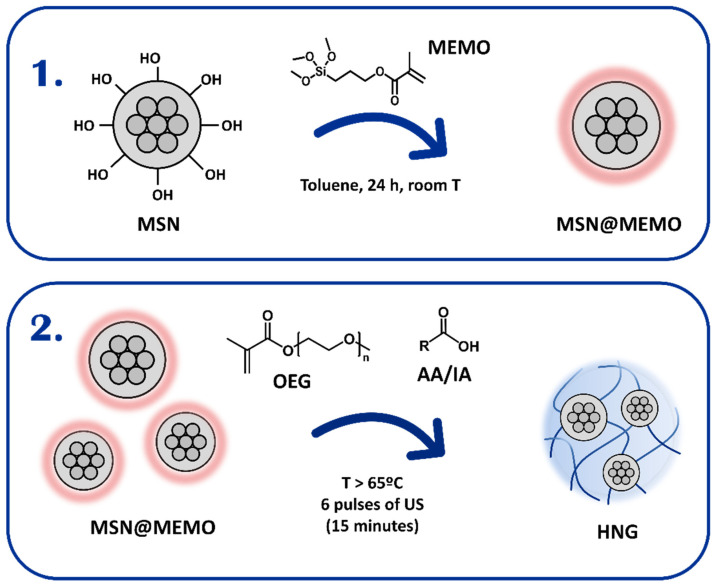
Scheme of the synthesis steps: surface silanization of MSNs (**1**) and polymerization (**2**) to produce HNGs.

**Figure 3 nanomaterials-12-03835-f003:**
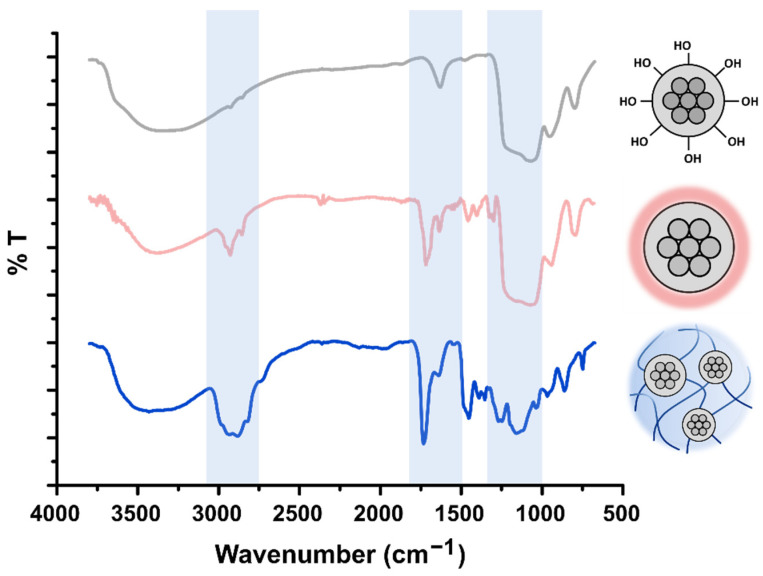
FT-IR spectra of pristine MSNs, functionalized MSNs (MSN@MEMO), and HNGs in water (HNG-P(DEGMA)).

**Figure 4 nanomaterials-12-03835-f004:**
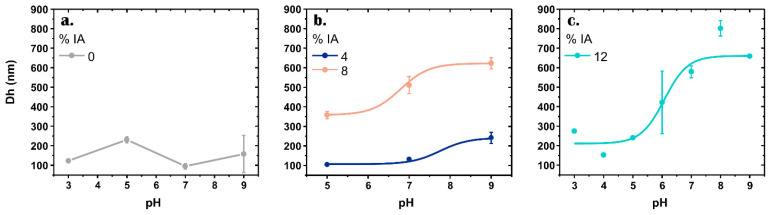
The effect of pH on the hydrodynamic particle diameter of: (**a**) HNG-P(DEGMA); (**b**) HNG-P(DEGMA-co-IA_4_) and HNG-P(DEGMA-co-IA_8_) and (**c**) HNG-P(DEGMA-co-IA_12_).

**Figure 5 nanomaterials-12-03835-f005:**
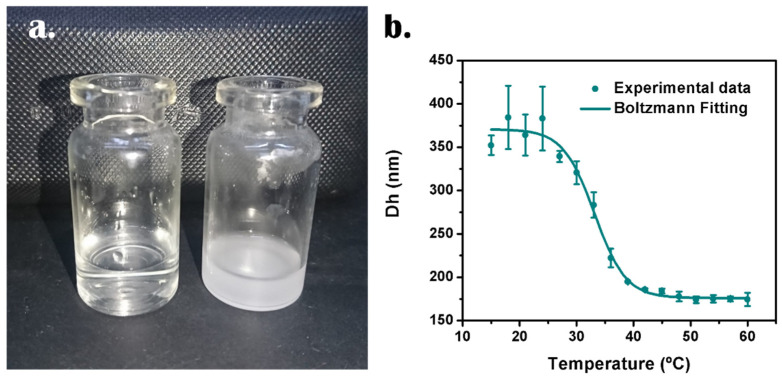
(**a**) Photograph representing the thermo-response of HNG-P(DEGMA-co-IA_12_): on the left, the suspension at room temperature and on the right, the same suspension at around 60 °C. (**b**) Average Dh vs. temperature of HNG-P(DEGMA-co-IA_12_).

**Figure 6 nanomaterials-12-03835-f006:**
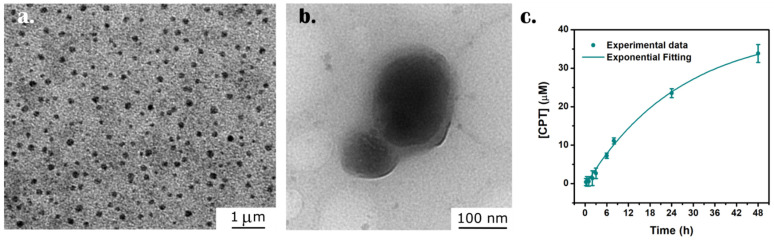
(**a**) Low magnification TEM image showing mostly monodispersed HNG-P(DEGMA-co-IA_12_) nanoparticles. (**b**) High magnification TEM image of an isolated HNG. (**c**) Release study in PBS pH 7.4 and 40 °C.

**Figure 7 nanomaterials-12-03835-f007:**
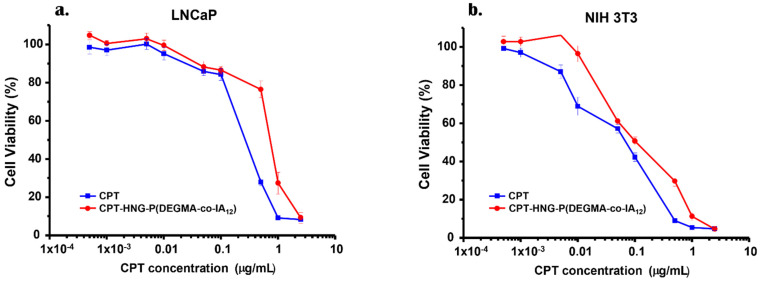
MTS cell viability assays in LNCaP (**a**) and NIH 3T3 (**b**) cell lines. Cell viability data are expressed as mean ± SEM (n = 3).

**Table 1 nanomaterials-12-03835-t001:** Summary of the reaction conditions of synthesized HNGs.

Samples	DEGMA:OEGMA (mmol)	MSN@MEMO (mg)	TEGDMA (%)	AA/IA (%)
HNG-P(DEGMA-co-OEGMA)^1.5^	DEGMA:OEGMA (0.8:0.2)	0.25	1.5	-
HNG-P(DEGMA-co-OEGMA)	DEGMA:OEGMA (0.8:0.2)	0.25	-	-
HNG-P(DEGMA)	DEGMA (1)	0.25	-	
HNG-P(DEGMA-co-AA_4_)	DEGMA (1)	0.25	-	4% AA
HNG-P(DEGMA-co-IA_4_)	DEGMA (1)	0.25	-	4% IA
HNG-P(DEGMA-co-IA_8_)	DEGMA (1)	0.25	-	8% IA
HNG-P(DEGMA-co-IA_12_)	DEGMA (1)	0.25	-	12% IA

**Table 2 nanomaterials-12-03835-t002:** HNGs synthesized starting from constant amount of MSN@MEMO DEGMA:OEGMA 80:20 with/without TEGDMA and AA.

8Samples	DEGMA: OEGMA (%)	TEGDMA (%)	AA (%)	Dh ± SD (nm) 25 °C	PDI 25 °C	VPTT (°C)
HNG-P(DEGMA-co-OEGMA)^1.5^	80:20	1.5	-	222.8 ± 2.8	0.202	59.5 ± 0.1
HNG-P(DEGMA-co-OEGMA)	80:20	-	-	177.8 ± 4.9	0.479	46.6 ± 0.3

**Table 3 nanomaterials-12-03835-t003:** HNGs synthesized HNGs synthesized starting from constant amount of MSN@MEMO and DEGMA with/without AA and IA.

Samples	Acidic Co-Monomer (%)	Dh ± SD (nm) 25 °C	PDI 25 °C	T_CP_ (°C)
HNG-P(DEGMA)	-	206.9 ± 103.6	0.142	23.6 ± 0.1
HNG-P(DEGMA-co-AA_4_)	4% AA	237.4 ± 27.7	0.210	34.6 ± 0.4
HNG-P(DEGMA-co-IA_4_)	4% IA	137.5 ± 40.3	0.186	28.00 ± 0.02

**Table 4 nanomaterials-12-03835-t004:** HNGs synthesized starting from constant amount of MSN@MEMO, DEGMA, and increasing percentage of IA.

Samples	Dh ± SD (nm) 25 °C	PDI 25 °C	ζ (mV)	Dh ± SD (nm) 50 °C	PDI 50 °C	T_CP_ (°C)
HNG-P(DEGMA)	206.9 ± 103.6	0.142	−1.5	600.2 ± 26.5	0.138	23.6 ± 0.1
HNG-P(DEGMA-co-IA_4_)	137.5 ± 40.3	0.186	−22.5	155.1 ± 6.5	0.048	28.00 ± 0.02
HNG-P(DEGMA-co-IA_8_)	238.1 ± 29.5	0.200	−28.5	179.7 ± 60.0	0.242	29.9 ± 0.2
HNG-P(DEGMA-co-IA_12_)	343.2 ± 56.4	0.388	−30.7	130.5 ± 11.1	0.260	37.2 ± 0.3

**Table 5 nanomaterials-12-03835-t005:** IC_50_ values of CPT and different CPT loaded HNG-P(DEGMA-co-IA_12_) obtained over LnCaP and NIH 3T3 cell lines.

Cell Lines	CPT	CPT Loaded HNG-P(DEGMA-co-IA_12_)
LNCaP	0.0670 ± 0.0101	0.0618 ± 0.0060
NIH 3T3	0.0476 ± 0.0039	0.0839 ± 0.0069

Each value indicates the mean ± SEM. All the experiments were carried out in triplicate. CPT or CPT equivalent expressed in μg/mL.

## Data Availability

Not applicable.

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
