# Peer review of "Mesoporous Silica and Oligo (Ethylene Glycol) Methacrylates-Based Dual-Responsive Hybrid Nanogels"

_nanomaterials, 2022, doi:10.3390/nano12213835_

Round 1

Author Response

The complete response to all queries is presented in a separated file.

Reviewer 2 Report

Macchione et al. describe the synthesis and in vitro testing of a silica/polymer nanoparticle for drug delivery. These particles combine several distinct concepts to arrive at a functional material that responds to heat and pH changes. The material is characterized throughout the synthetic process primarily by DLS, however the in vitro characterization needs work. I have included specific comments below in no particular order. 

The authors should define what a lower critical solution temperature and volume phase transition temperature are in the introduction. It would also be instructive to include the temperature range for the PEG LCST

A schematic or chemical structure of the polymer backbone above/below the LCST would significantly help the visualization of this transition. 

Measurements should be performed on the TEM images for comparison between the core and the hydrodynamic diameter. 

The authors need to explain the remarkably low standard deviation in table 2 in comparison with the PDI values, which indicate a significantly greater distribution. 

"From the table it can be seen that all the samples show a size distribution in the nanometric scale with a polydispersity index (PDI) values between 0.1 and 0.4, values which indicate monodispersed distribution of particles." This statement requires a reference. FDA typically considers PDI under 0.15 as monodisperse

Authors should state clearly the temperatures that are photographed in Figure 4A

Differential scanning calorimetry should be used to confirm the LCST of the construct and verify the transition temperature. 

Figure 4C is not instructive. There needs to be a significant number of particles shown to demonstrate that this is not an isolated phenomenon. Further, a quantification of the distribution of monomers, dimers, trimers, and multimers would augment the authors point. 

CPT release study should be performed at 37 C, particularly as the authors do not propose a way to raise the body temperature to 40 C. The data would also be significantly enhanced if this was compared to release at 25 C. Additionally, tumors are more acidic that normal tissue so release of CPT in a more environmentally accurate pH is warranted. Finally, CPT is highly hydrophobic and many components in serum are designed to bind and transport these hydrophobic materials, performing a release study with 10% FBS, ala cell culture media, would greatly enhance the validity of the results. It would also be instructive to know what percentage of the drug is released over the 48 h, although I do appreciate 

The kind of cells (fibroblast, prostate ca) should be reiterated in the results. The comparison of human tumor cells with mouse fibroblasts is inappropriate. The same species should be used. 

The authors offer no evidence of cell uptake and their release data strongly suggest that the CPT can release in culture media without uptake. They need to either prove cellular uptake directly or they need to include some sort of controls that either prevent uptake and therefore maintain viability.

Author Response

(The authors gave the same response as above.)

Reviewer 3 Report

The authors evaluated the mesoporous silica and oligo (ethylene glycol) methacrylates-based dual-responsive hybrid nanogels. This is an interesting article using organic and inorganic components for nanomaterials preparation. But the authors didn’t evaluate the safety, efficacy, and pharmacokinetics of these nanoparticles in preclinical models.

Below are the comments and provide the information in the manuscript:

1.     Introduction is long. It should be reduced. It doesn’t discuss why the authors used the Camptothecin drug for nanomaterials preparation.

2.     The authors should include the source location for reagents and chemicals in the 2.1 section.

3.     Should include the institutional animal ethics committee IAEC number in the 2.3 section. In the 2.1 section, subculturing and trypsinization procedures need to be included.

4.     Why the authors used only camptothecin drug for these nanoparticles? The authors used the UV-visible spectrometry method for drug analysis. How was this method validated? These results need to be mentioned. Why didn’t the authors use the liquid chromatography method for quantification and validation?

5.     For nanoparticle characterization, the authors didn’t evaluate the viscosity, long-term stability, or drug-excipient compatibility.

6.     In figure 4, the authors need to replace the TEM images with clear representation. The drug release profile shows only 30-40 % release. The authors need to show the complete release profile with release rate characteristics.

7.     In table 5, the units for IC50 values should be mentioned. How the authors calculated the IC50 values from the cytotoxicity profiles? the ic50 values should be checked again.

8.     The authors need to show the data of safety, efficacy, and pharmacokinetics of Camptothecin from nanoparticles in comparison with pure drug.

Author Response

(The authors gave the same response as above.)

Round 2

Reviewer 3 Report

The authors significantly improved the manuscript and answered all the comments.